# Evaluation Study of xMAP TIER Assay on a Microsphere-Based Platform for Detecting First-Line Anti-Tuberculosis Drug Resistance

**DOI:** 10.3390/ijerph192417068

**Published:** 2022-12-19

**Authors:** Xichao Ou, Zhiguo Zhang, Bing Zhao, Zexuan Song, Shengfen Wang, Wencong He, Shaojun Pei, Dongxin Liu, Ruida Xing, Hui Xia, Yanlin Zhao

**Affiliations:** 1National Center for Tuberculosis Control and Prevention, Chinese Center for Disease Control and Prevention, Beijing 102206, China; 2Tuberculosis Dispensary of Changping District, Beijing 102202, China; 3School of Public Health, Peking University, Beijing 100191, China

**Keywords:** *Mycobacteirum tuberculosis*, diagnosis, xMAP, drug resistance

## Abstract

Early diagnosis of drug susceptibility for tuberculosis (TB) patients could guide the timely initiation of effective treatment. We evaluated a novel multiplex xMAP TIER (Tuberculosis-Isoniazid-Ethambutol-Rifampicin) assay based on the Luminex xMAP system to detect first-line anti-tuberculous drug resistance. Deoxyribonucleic acid samples from 353 Mycobacterium tuberculosis clinical isolates were amplified by multiplex polymerase chain reaction, followed by hybridization and analysis through the xMAP system. Compared with the broth microdilution method, the sensitivity and specificity of the xMAP TIER assay for detecting resistance was 94.9% (95%CI, 90.0–99.8%) and 98.9% (95%CI, 97.7–100.0%) for rifampicin; 89.1% (95%CI, 83.9–94.3%) and 100.0% (95%CI, 100.0–100.0%) for isoniazid; 82.1% (95% CI, 68.0–96.3%) and 99.7% (95% CI, 99.0–100.0%) for ethambutol. With DNA sequencing as the reference standard, the sensitivity and specificity of xMAP TIER for detecting resistance were 95.0% (95% CI, 90.2–99.8%) and 99.6% (95% CI, 98.9–100.0%) for rifampicin; 96.9% (95% CI, 93.8–99.9%) and 100.0% (95% CI, 100.0–100.0%) for isoniazid; 86.1% (95% CI, 74.8–97.4%) and 100.0% (95% CI, 100.0–100.0%) for ethambutol. The results achieved showed that the xMAP TIER assay had good performance for detecting first-line anti-tuberculosis drug resistance, and it has the potential to diagnose drug-resistant tuberculosis more accurately due to the addition of more optimal design primers and probes on open architecture xMAP system.

## 1. Introduction

Drug-resistant (DR) tuberculosis (TB) is an increasing threat to public health worldwide; the initially diagnosed TB patients who had multidrug-resistant (MDR) TB/rifampicin-resistant (RR) TB has remained at about 3–4% and, for previously treated TB patients, it has remained at about 18–21% [1,2]. The WHO End TB Strategy requires drug susceptibility testing (DST) of all people with TB but there are still large gaps in the detection of DR-TB. In 2020, 30% of people diagnosed with bacteriologically confirmed pulmonary TB were not tested for anti-tuberculosis drug resistance [1,2], and resistance detection to the three most effective first-line drugs; rifampicin (RIF), isoniazid (INH) and ethambutol (EMB) is of greatest concern [1]. 

Conventional DST by the culture method is commonly used for DR TB detection; however, these methods are time-consuming and laborious. Therefore, the conventional DST cannot provide the timely diagnosis required for proper patient treatment; in addition, these methods pose safety threats to the personnel. Most *Mycobacterium tuberculosis (M. tuberculosis)* isolates with resistance to RIF, INH, and EMB have mutations at specific codons in a small number of relative genes. RIF resistance involves mutations in the *rpoB* gene [3], INH resistance is mostly associated with mutations in *inhA* and *katG* genes [4], and resistance to EMB involves mutations in *embB* [5]. Therefore, *M. tuberculosis* can also be tested for drug resistance using alternative rapid and safer genotypic assays, which detect resistance by identifying mutations known to confer resistance to first-line anti-tuberculosis drugs mentioned above [6,7,8,9]. During the past decades, several efficient molecular assays have been recommended by the WHO for the timely and accurate diagnosis of DR TB, including Abbott RealTime MTB RIF/INH, BD MAX MDR-TB, Cobas MTB-RIF/INH, FluoroType MTBDR, and Xpert MTB/RIF which was also approved by US Food and Drug Administration (FDA) [10].

The Luminex MagPlex microsphere-based multi-analyte profiling (xMAP) platform (Luminex Corp. Austin, TX, USA) is a multiplexed detection system capable of analyzing and reporting up to 500 different targets in a single reaction vessel [11]. By using labeled microspheres or beads, which allow for the simultaneous capture of multiple analytes from a single reaction, xMAP offered a new platform for high-throughput DR-TB diagnoses through the integration of multiplex PCR amplification and microsphere hybridization technology. In this study, we report a novel multiplex TIER (Tuberculosis-Isoniazid-Ethambutol-Rifampicin) assay using the xMAP platform for the detection of three first-line anti-tuberculosis drug resistance, and its performance evaluation compared with phenotypic broth microdilution DST and whole-genome sequencing.

## 2. Materials and Methods

### 2.1. Mycobacteria tuberculosis Strain Collection and Culture 

*Mycobacteria tuberculosis* clinical isolates were selected from the National Tuberculosis Reference Laboratory in China; 353 strains isolated from pulmonary tuberculosis patients were sub-cultured on Löwenstein–Jensen medium and incubated at 37 °C for 3–4 weeks.

### 2.2. Phenotypic Broth Microdilution Drug Susceptibility Test

Phenotypic DST was performed using the broth microdilution method which provides a minimal inhibitory concentration (MIC) value (Sensititre^®^ MYCOTB plate). A panel of 13 anti-TB drugs was tested, including INH, RPF, and EMB. Several colonies were selected from growth on solid media using a sterile loop and inoculated into a test tube containing glass beads as described previously [12]. A 0.5 McFarland standard equivalent was made after vortexing, suspensions were diluted 100-fold with the addition of 100 μL of the 0.5 Mc suspensions to 10 mL of Mueller–Hinton broth with OADC (oleic acid-albumin-dextrose-catalase). Aliquots of 100 μL of the standard 1.5 × 10^5^ CFU/mL inoculum were distributed to each well using the semiautomated SensititreTM Auto-inoculator (Thermo Fisher, Scientific Inc., San Francisco, CA, USA). MYCOTB plates were sealed and incubated at 37 °C. MIC was defined as the lowest concentration without obvious visible bacterial growth compared with positive controls, aided by the VizionTM Digital viewing system. Mtb H37Rv (ATCC 27294) was used as a quality control in each batch. Based on CLSI M62, the breakpoints for INH, RPF, and EMB were 0.12 μg/mL, 1 μg/mL and 2 μg/mL, respectively. An inconclusive result for ethambutol was an MIC of 4 μg/mL, which does not correlate with either a susceptible or resistant result.

### 2.3. xMAP TIER Assay

For each isolate, one loop of cultured colonies from Löwenstein–Jensen culture medium was suspended in 500 μL of TE buffer and boiled at 100 °C for 15 min. After cooling, the suspension was centrifuged at 12,000× *g* for 5 min and 200 μL of the supernatant containing the nucleic acid was transferred into a new microcentrifuge tube for the xMAP TIER assay (Hygienland Shanghai Biotechnology Co., Ltd., Shanghai, China). This assay is equipped with two different internal controls (16S, λ bacteriophages) for quality control throughout the whole detection process, including nucleic acid extraction, multiplex PCR amplification, and hybridization. The main steps for the xMAP TIER assay are shown in Figure 1. Several primers were synthesized to amplify gene fragments harboring resistance mutations in *rpoB*, *katG*, *inhA*, and *embB* genes; seventeen specific mutations at 10 mutation sites can be detected by this assay (Table 1). PCR was performed in a 25 μL reaction mixture containing 1 μL of template DNA, 1 μL λDNA, and 23 μL PCR master mix with the following thermal profile: initial denaturation at 95 °C for 5 min followed by 20 repeated cycles of melting at 95 °C for 30 s, annealing at 65 °C for 90 s, extension at 72 °C for 30 s, respectively; a second round of PCR was performed with 20 cycles of denaturation at 95 °C for 30 s, annealing at 56 °C for 90 s, extension at 72 °C for 30 s, respectively; an additional incubation for 10 min at 68 °C, and then cooled to 4 °C. Microsphere probe hybridization was carried out with 33 μL 1.5× tetramethylammonium chloride (TMAC) hybridization buffer mixed with 6 μL hybrid microspheres and 12 μL Tris-EDTA buffer (pH 8.0) for each reaction, five microliters of PCR product were then added, the mixture was denatured at 95 °C for 5 min and hybridized at 63 °C for 15 min. Ten microliters of streptavidin-R-phycoerythrin (50 μg/mL) were added for further incubation at 63 °C for 10 min, the results were read immediately using the MAGPIX (Luminex Corp., Austin, TX, USA) System. The cut-off value for a positive result was set at ≥1.5 times the median fluorescence intensity (MFI) values of the negative control by the manufacturer. 

### 2.4. Whole Genome Sequencing (WGS)

Genomic DNA of MTB isolates prepared by the cetyltrimethylammonium bromide extraction method was subjected to WGS using the Illumina HiSeq 2000 platform as described previously [13,14]. The Clockwork pipeline was used to process all raw WGS data [15]. Compared with Mtb H37Rv strain (NC_000962.3), mutations were called when they were present in more than two reads in the forward and reverse directions. Mutations in genes of the proline–glutamic acid (PE)/proline–proline–glutamic acid (PPE) family and in regions with repetitive sequences were excluded. WGS predictions of phenotypic susceptibility were based on the WHO-endorsed catalog of mutations detected in or upstream of drug-resistance-related genes [16]. Isolates with resistance-conferring mutations ranked in Group 1 (associated with resistance) and Group 2 (associated with resistance-interim) were considered to be genotype drug-resistant [16].

### 2.5. Statistical Analysis

Compared with phenotypic DST and WGS, the sensitivity and specificity of the xMAP TIER assay for detecting RIF, INH and EMB resistance were calculated with an exact Clopper–Pearsong confidence interval. 

## 3. Results 

A total of 353 MTB isolates from NTRL were collected for validation. Drug susceptibility testing for RIF, INH and EMB was performed using xMAP TIER assay, broth microdilution MIC method and WGS, respectively. On the basis of the broth microdilution DST method, 61 isolates were MDR, 77 isolates were INH mono-resistant, 17 isolates were RIF mono-resistant, and 198 isolates were INH- and RIF-sensitive (Figure 2). Inconclusive results for EMB resistance detection occurred for 14 isolates due to an MIC of 4 μg/mL. Of the 339 MTB isolates which had conclusive EMB susceptibility test results by broth microdilution, 28 (8.3%) were resistant to EMB. Among the 353 MTB isolates, three main lineages were identified: 66.2% (234/353) were assigned to lineage 2 (East Asian genotype), 5.6% (20/353) to lineage 3 (India and East Africa genotype), and 28.2% (99/353) to lineage 4 (Euro-American genotype) (Figure 2). Isolates with detectable mutations in *rpoB*, *katG*, *inhA*, *ahpC* and *embB* genes accounted for 22.4% (79/353), 30.0% (106/353), 8.5% (30/353), 1.1% (4/353), and 10.5% (37/353), respectively (Figure 2).

Compared with the broth microdilution MIC method, the sensitivity and specificity of the xMAP TIER assay for RIF resistance detection were 94.9% (95%CI, 90.0–99.8%) and 98.9% (95%CI, 97.7–100.0%), respectively. The sensitivity and specificity for INH resistance detection were 89.1% (95% CI, 83.9–94.3%) and 100.0% (95% CI, 100.0–100.0%), respectively. The sensitivity and specificity for detecting EMB resistance were 82.1% (95% CI, 68.0–96.3%) and 99.7% (95% CI, 99.0–100.0%), respectively (Table 2).

Of the 78 isolates with a phenotypic RIF-resistant result, four isolates were identified as RIF susceptible by the xMAP TIER assay. WGS results indicated these four isolates contained genotypic mutations (Q432E, H445L, H445R) not included in the xMAP TIER assay. Three phenotypic RIF susceptible isolates were identified as RIF resistant by the xMAP TIER assay and WGS analysis revealed that two of these were confirmed to be genotypically resistant due to a mutation (L430P) in the *rpoB* gene (Table 2). In the third strain, an S450L mutation was detected by the xMAP TIER assay but no mutation was found by WGS. 

A consistent INH susceptibility result was found for 338 isolates using the xMAP TIER assay and phenotypic DST. A total of 15 phenotypically INH-resistant isolates were tested as INH susceptible by the xMAP TIER assay. WGS revealed that five of these carried no mutations, which may be due to unknown resistance mechanisms. The remaining 10 isolates contained genotypic mutations not included in the xMAP TIER assay (Table 2). For three isolates, a *katG*_S315N mutation was found which is listed in Group 1 by the WHO catalog, and a *katG*_2050_ins_2_T_TGC was found in one discordant strain which was listed in Group 2. The other six isolates had mutations not listed in Group1 or Group2 (*katG*_Asn138Asp; *katG*_Val1Ala; *ahpC*_-52C>T, *katG*_Asp735Ala; *ahpC*_c-81t; *ahpC*_-48G>A, *katG*_ Asp94Gly).

Fourteen isolates gave inconclusive MIC DST results for EMB. Out of the 339 isolates with conclusive EMB susceptibility results, five phenotypically EMB-resistant isolates were called EMB susceptible by the xMAP TIER assay. Of these, four isolates had genotypic mutations (*embA*_C-12T, *embB*_Y319C; *embB*_Y319S; *embB*_G406A; *embB*_G406S) outside of the detection scope of xMAP TIER assay. No mutation was found in the fifth strain, perhaps due to unknown resistance mechanisms. One strain was determined to be EMB resistant by the xMAP TIER assay but susceptible to EMB by MIC DST and had a mutation in *embB* that was detected by WGS (*embB*_M306V).

With WGS as the reference standard, mutations ranked in Group 1 and Group 2 were considered to be genotypically drug-resistant. The sensitivity and specificity of xMAP TIER detection of resistance were 95.0% (95% CI, 90.2–99.8%) and 99.6% (95% CI, 98.9–100.0%) for rifampicin; 96.9% (95% CI, 93.8–99.9%) and 100.0% (95% CI, 100.0–100.0%) for isoniazid; 86.1% (95% CI, 74.8–97.4%) and 100.0% (95% CI, 100.0–100.0%) for ethambutol (Table 3). Five isolates were EMB resistant by WGS but classified as EMB susceptible by the xMAP TIER assay. Of these, four isolates contained genotypic mutations (*embA*_C-12T, *embB*_Y319C; *embB*_Y319S; *embB*_G406A; *embB*_G406S) outside of the detection scope of the xMAP TIER assay. The fifth strain contained an *embB*_G406D which was not included in the xMAP TIER assay but was also excluded from further analysis due to an inconclusive MIC DST result for EMB.

## 4. Discussion

Early diagnosis and treatment are crucial to prevent the transmission of drug-resistant *M. tuberculosis* [17]. Several new molecular diagnostics have revolutionized the diagnosis of drug-resistant tuberculosis. Multiplexing technologies that allow for the detection of multiple nucleic acid sequences in a single reaction, can greatly reduce the time, cost, and labor. The Luminex xMAP system is a multiplexed microsphere-based platform with an open-architecture design and has been used to develop various assays for clinical disease diagnosis [18,19], such as the xTAG gastrointestinal pathogen panel and NxTAG respiratory pathogen panel has been widely used and approved by FDA. We evaluated a novel xMAP TIER assay to detect three first-line anti-tuberculous drug resistance. The xMAP TIER assay integrates multiplex PCR amplification and Microsphere hybridization, enabling the detection of all single nucleotide polymorphism (SNP) types in a single tube. Our results from this preliminary evaluation showed that the TIER assay can timely and effectively diagnose isoniazid, ethambutol, and rifampicin-resistant isolates. Compared with the phenotypic broth microdilution DST, the turnaround time of xMAP TIER was greatly shortened from 10 days to 5 h. With sequencing as the reference standard, the WHO has set targets for diagnostic sensitivity and specificity of next-generation molecular DST assays [20]. The xMAP TIER assay reached the sensitivity target (95%) for RIF and INH but missed the sensitivity target for EMB, and it met the specificity target (98%) for all three first-line anti-tuberculous drugs.

Recent studies have shown that patients infected with *M. tuberculosis* with borderline *rpoB* mutations were misclassified as sensitive by phenotypic drug susceptibility testing and often fail treatment due to low-level resistance to RIF [21,22]. Leu452Pro or Leu430Pro mutations of *rpoB* in rifampicin-resistant isolates were more likely to be missed by phenotypic DST [23]. Three isolates of RIF susceptible to broth microdilution DST were identified as RIF resistant by the xMAP TIER assay in this study, but WGS analysis revealed that two of these were confirmed to be genotypically resistant due to a Leu430Pro mutation in the *rpoB* gene; both had MICs of 0.5 μg/mL. A Leu452Pro mutation was detected by the xMAP TIER assay in the third strain, but no mutation was found by WGS, which might be due to heteroresistance. The application of molecular assays for the initial diagnosis of tuberculosis will allow for the detection of more cases that are falsely susceptible to RIF by phenotypic DST. WGS found four isolates with genotypic mutations (Q432E, H445L and H445R) not included in the xMAP TIER assay, indicating that the probes for RIF resistance detection in this assay still need modification and optimization.

The TIER assay interrogates drug resistance by mutations in the *katG* and *inhA* genes which are responsible for about 75% of isoniazid-resistant isolates [24]. The sensitivity of the TIER assay for the detection of INH resistance was 89.1%. Fifteen phenotypically INH-resistant isolates were classified as INH susceptible by the xMAP TIER assay. WGS revealed that 11 isolates carried no mutations in *katG* or *inhA*, which may be due to resistance-associated mutations outside of the *katG* and *inhA* genes or other unknown resistance mechanisms. Four isolates were found to have genotypic mutations (*katG*_S315N, *katG*_2050_ins_2_T_TGC) not included in the xMAP TIER assay.

In our analysis, EMB resistance of the TIER assay was detected with a sensitivity of 82.1%. Low performance of EMB resistance detection was also observed in other studies [25,26] and could be due to some mutation-containing codons that were not included in the TIER assay. The development of the WHO catalog of MTBC mutations associated with drug resistance has provided a global standard for resistance interpretation and can improve the design of molecular diagnostics. Further analysis of the isolates for which the TIER assay failed to detect resistance mutations provided insights for further optimization. Importantly, the TIER assay did not miss the detection of any of the mutations that were included in the assay. The open architecture of the xMAP system allows the addition of more specific primers and probes, so the TIER assay has the potential of detecting more resistance-conferring mutations, and strain susceptibility to more anti-Tuberculosis drugs. 

This was just a preliminary validation study of a new molecular assay based on the xMAP system for detecting RIF, INH and EMB resistance, and just MTB isolates were tested for evaluation. The preliminary validation performance looks good and with the assay updated based on the results of this study, more MTB isolates and clinical samples, such as sputum, will be recruited for a more extensive evaluation study in the future.

## 5. Conclusions

In conclusion, the TIER assay showed good sensitivity in detecting RIF, INH, and EMB resistance. The inclusion of additional primers and probes to improve the sensitivity of drug resistance detection would enhance the use of the TIER assay in future clinical studies.

## Figures and Tables

**Figure 1 ijerph-19-17068-f001:**
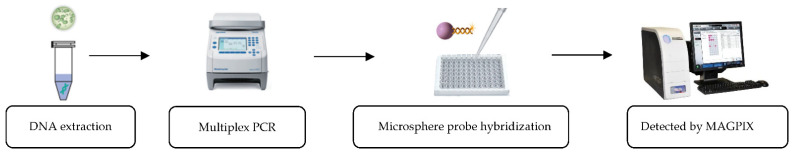
xMAP TIER test workflow.

**Figure 2 ijerph-19-17068-f002:**
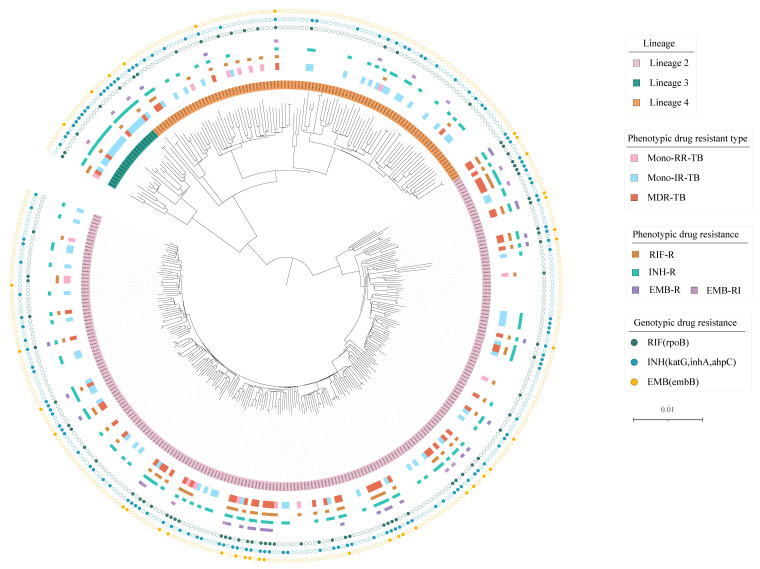
Maximum-likelihood tree of 353 MTB isolates and annotated with drug-resistance information. Note: Lineages, phenotypic drug-resistant type, and genotypic drug-resistant profile of isolates are shown.

**Table 1 ijerph-19-17068-t001:** Specific mutations designed for detection in the xMAP TIER assay.

Drug	Gene	Wild Type	Mutation
INH			
	*katG*	315AGC	315ACC
	*inhA*	-777C	-777T
		-770T	-770C
RIF			
	*rpoB*	430CTG	430CCG
	*rpoB*	432CAA	432AAA or 432CAA
	*rpoB*	435GAC	435GTC or 435GGC
	*rpoB*	445CAC	445TAC or 445GAC
	*rpoB*	450TCG	450TTG
	*rpoB*	452CTG	152CCG
EMB	*embB*	306ATG	306GTG, 306CTG, 306ATA, 306ATT or 306ATC

**Table 2 ijerph-19-17068-t002:** Diagnostic performance of the xMAP TIER assay compared with the phenotypic DST method for detection of RIF, INH, and EMB susceptibility.

Drugs	Phenotypically DST Resistant			Phenotypically DST Susceptible			Sensitivity (95%CI)	Specificity (95%CI)
	xMAP Resistant	xMAP Susceptible	Total	xMAP Resistant	xMAP Susceptible	Total		
Rifampicin	74	4 ^a^	78	3 ^b^	272	275	94.9 (87.4–98.6)	98.9 (96.9–99.8)
Isoniazid	123	15 ^c^	138	0	215	215	89.1 (82.7–93.8)	100.0 (98.3–100.0)
Ethambutol	23	5 ^d^	28	1 ^e^	310	311	82.1 (63.1–93.9)	99.7 (98.2–100.0)

^a^ Four isolates contained genotypic mutation in *rpoB* by WGS. ^b^ Two isolates showed genotypic mutation in *rpoB* by sequencing, the third discordant strain has no mutation according to WGS. ^c^ Ten isolates contained genotypic mutation by WGS. ^d^ Four isolates have mutations in *embB* according to WGS. ^e^ Mutation in *embB* has been detected by WGS.

**Table 3 ijerph-19-17068-t003:** Diagnostic performance of the xMAP TIER assay compared with WGS for detection of RIF, INH, and EMB susceptibility.

Drugs	Genotypically Resistant by WGS			Genotypically Susceptible by WGS			Sensitivity (95%CI)	Specificity (95%CI)
	xMAP Resistant	xMAP Susceptible	Total	xMAP Resistant	xMAP Susceptible	Total		
Rifampicin	76	4	80	1	272	273	95.0 (87.7–98.6)	99.6 (98.0–100.0)
Isoniazid	123	4	127	0	226	226	96.9 (93.8–99.9)	100.0 (100.0–100.0)
Ethambutol	31	5	36	0	317	317	86.1 (70.5–95.3)	100.0 (98.8–100.0)

## Data Availability

The data presented in this study are available in Appendix A.

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
