# Peer review of "Evaluation Study of xMAP TIER Assay on a Microsphere-Based Platform for Detecting First-Line Anti-Tuberculosis Drug Resistance"

_ijerph, 2022, doi:10.3390/ijerph192417068_

Round 1
Reviewer 1 Report
Ou et al. evaluated a novel multiplex xMAP TIER assay on the Luminex xMAP system to detect first-line anti-TB drug resistance using 353 MTB clinical isolates by comparing with broth microdilution method and DNA sequencing through WGS. High sensitivity and specificity of xMAP TIER for detecting RIF and INH. Relatively low sensitivity but high specificity for detecting EMB resistance were observed. The study was performed well, and the results are very informative. If the authors can address some comments below would improve the manuscript.
1. Line 55-56: Please provide information regarding which of these mentioned test(s) is/are US FDA approved and which is/are approved by the European Medicines Agency (EMA) for IVD use.
2. Chang strains to isolates throughout the manuscript.
3. Please clarify if WGS results indicated any mutation in rpoB, inhA, katG, and embB in MTB isolates, always consider those isolates are drug resistant. Listing what WHO defines what mutations in those genes predict drug resistance/treatment failure will be helpful.
4. Line 167: should it be 353 instead of 340?
5. Line 177: should it be 339 instead of 341?
6. Table 2: Row of isoniazid: should it be 138 instead of 150? Row of ethambutol, should it be 311 instead of 313?
7. Line 198-199: I don’t see any isolate was excluded from the study for xMAP while comparing with WGS because the total number is still 353 in table 3 row Ethambutol (36+317=353). If this is true, why put this sentence (line 198-199) here?
8. Table 3, label “Genotypically resistant” to “Genotypically resistant by WGS”. Same to susceptible.
9. Line 208-210: Any assays that use this technology/system are US FDA approved? If yes, give some examples. If not, do you know why still not? What are the hurdles?
10. What do borderline rpoB mutations mean?
11. Line 253: remove the sentence “We also realize the limitations of this study.” Instead, clearly state this study's limitations in this paragraph.
Author Response
Dear reviewer,
Thank you very much for the comments from you. Based on the comments, we have revised the relevant part in the original manuscript. A document answering every question was also summarized and enclosed below. We hope that the revised manuscript is acceptable for further review.
Thank you very much for your continued attention.
1) Line 55-56: Please provide information regarding which of these mentioned test(s) is/are US FDA approved and which is/are approved by the European Medicines Agency (EMA) for IVD use.
Answer: Thanks for your comment. The Xpert MTB/RIF was also approved by US Food and Drug Administration (FDA), which has added in the manuscript.
2) Chang strains to isolates throughout the manuscript.
Answer: Thanks for the suggestion, the modification has been performed accordingly.
3) Please clarify if WGS results indicated any mutation in rpoB, inhA, katG, and embB in MTB isolates, always consider those isolates are drug resistant. Listing what WHO defines what mutations in those genes predict drug resistance/treatment failure will be helpful.
Answer: Thanks for your reminding and we have revised the relevant part to make it easier to understand.
4) Line 167: should it be 353 instead of 340?
Answer: Sorry for the clerical error, it should be 338 instead of 340.
5) Line 177: should it be 339 instead of 341?
Answer: Thanks for your carefully review, it has been revised.
6) Table 2: Row of isoniazid: should it be 138 instead of 150? Row of ethambutol, should it be 311 instead of 313?
Answer: Many thanks, these number has been rectified.
7) Line 198-199: I don’t see any isolate was excluded from the study for xMAP while comparing with WGS because the total number is still 353 in table 3 row Ethambutol (36+317=353). If this is true, why put this sentence (line 198-199) here?
Answer: Yes, no isolate was excluded from the study for xMAP while comparing with WGS. The sentence (line 198-199) put here was aiming to further analyze the limitation of xMAP test in the discussion part.
8) Table 3, label “Genotypically resistant” to “Genotypically resistant by WGS”. Same to susceptible.
Answer: Thanks for your suggestion, the modification has been performed accordingly.
9) Line 208-210: Any assays that use this technology/system are US FDA approved? If yes, give some examples. If not, do you know why still not? What are the hurdles?
Answer: Thanks for your suggestion, and we have added two assays that approved by FDA in the manuscript.
10) What do borderline rpoB mutations mean?
Answer: Inappropriately high breakpoints by phenotypic drug susceptibility testing methods may exacerbate the rate of misclassification of rpoB mutations. A patient infected with a strain with borderline rpoB mutation might be classified as rifampicin-susceptible, followed by standard rifampicin-based therapy, and then eventually treatment failure or relapse. The relevant part has been modified.
11) Line 253: remove the sentence “We also realize the limitations of this study.” Instead, clearly state this study's limitations in this paragraph.
Answer: Thanks for your comment, this part has been rephrased.

Reviewer 2 Report
Thanks for the opportunity to review this manuscript.
The paper is well written and it has an appropriate design to answer the scientific question.
I have few minor comments:
1. In the methods section they authors state that the clinical isolates were selected from the National Tuberculosis Reference Laboratory in China. Could the authors elaborate from what type of sample these clinical isolates come from or from what type of tuberculosis cases they correspond to (for example, pulmonary tuberculosis)? If this is not possible, please clarify it in the methods section.
2. It seems that the authors are computing the 95% confidence interval (CI) for the proportions (sensitivity and specificity) using the normal approximation. This is generally correct, but when the proportion is very close to 1, the upper limit of the 95% CI could be above 1. It seems that this happened in some cases and the authors only truncated the upper limit in 1 (or 100%). This is not correct since the interval is no longer a 95% CI. See Agresti A, Caffo B. Simple and effective confidence intervals for proportions and differences of proportions result from adding two successes and two failures. The American Statistician. 2000 Nov 1;54(4):280-8, for an alternative method to compute 95% CIs when the proportion is close to 1.
3. Please double check the counts in Table 2. For example, in the phenotypically isoniazid DST resistant group, there are 123 xMAP resistant and 15 xMAP susceptible, and a total of 150, but 123 + 15 = 138. Similar case in the phenotypically ethambutol DST susceptible group.
4. Figure 2 is hard to read, please increase resolution.
Author Response
Dear reviewer,
Thank you very much for the comments from you. Based on the comments, we have revised the relevant part in the original manuscript. A document answering every question was also summarized and enclosed below. We hope that the revised manuscript is acceptable for further review.
Thank you very much for your continued attention.
1) In the methods section the authors state that the clinical isolates were selected from the National Tuberculosis Reference Laboratory in China. Could the authors elaborate from what type of sample these clinical isolates come from or from what type of tuberculosis cases they correspond to (for example, pulmonary tuberculosis)? If this is not possible, please clarify it in the methods section.
Answer: Yes, you are right. The modification has been performed accordingly.
2) It seems that the authors are computing the 95% confidence interval (CI) for the proportions (sensitivity and specificity) using the normal approximation. This is generally correct, but when the proportion is very close to 1, the upper limit of the 95% CI could be above 1. It seems that this happened in some cases and the authors only truncated the upper limit in 1 (or 100%). This is not correct since the interval is no longer a 95% CI. See Agresti A, Caffo B. Simple and effective confidence intervals for proportions and differences of proportions result from adding two successes and two failures. The American Statistician. 2000 Nov 1;54(4):280-8, for an alternative method to compute 95% CIs when the proportion is close to 1.
Answer: Thanks for your professional suggestion, the number has been corrected in the manuscript.
3) Please double check the counts in Table 2. For example, in the phenotypically isoniazid DST resistant group, there are 123 xMAP resistant and 15 xMAP susceptible, and a total of 150, but 123 + 15 = 138. Similar case in the phenotypically ethambutol DST susceptible group.
Answer: Thanks for your carefully review, the counts in Table 2 has been double check and revised.
4) Figure 2 is hard to read, please increase resolution.
Answer: Thanks for your comments, the Figure has been inserted in the manuscript and the single TIF image document has been uploaded.
